# HOXA9 Transcriptionally Promotes Apoptosis and Represses Autophagy by Targeting NF-κB in Cutaneous Squamous Cell Carcinoma

**DOI:** 10.3390/cells8111360

**Published:** 2019-10-31

**Authors:** Shuo Han, Xue Li, Xiaoting Liang, Liang Zhou

**Affiliations:** Guangdong Provincial Key Laboratory of Tropical Disease Research, Department of Toxicology, School of Public Health, Southern Medical University, Guangdong 510515, Guangzhou, China; hanshuo@i.smu.edu.cn (S.H.); lixue1234567@i.smu.edu.cn (X.L.); liangxiaoting658@i.smu.edu.cn (X.L.)

**Keywords:** HOXA9, NF-κB, apoptosis, autophagy, cutaneous squamous cell carcinoma

## Abstract

Tumor suppressor HOXA9 has been identified to promote apoptosis in cutaneous squamous cell carcinoma (cSCC). However, the mechanism of such pro-apoptotic role of HOXA9 remains obscure. KEGG (Kyoto Encyclopedia of Genes and Genomes) analysis of RNA-seq data showed that NF-κB, apoptosis and autophagy pathways are significantly regulated after HOXA9 knockdown. HOXA9 transcriptionally regulates RELA, the p65 subunit of NF-κB. Loss of HOXA9 in cSCC significantly upregulates RELA expression and thus enhances NF-κB pathway. Interestingly, RELA transcriptionally promotes not only anti-apoptotic factor BCL-XL but also autophagic genes including ATG1, ATG3, and ATG12. Our results reveal an enhanced NF-κB signaling network regulated by HOXA9, which contributes to repressed apoptosis and activated autophagy in cSCC development and may represent an intervention target for cSCC therapy.

## 1. Introduction

Cutaneous squamous cell carcinoma (cSCC) is the second most common type of cancer with an annual incidence of over one million globally [1,2,3]. Chronic ultraviolet (UV) exposure was suggested to be responsible for DNA damage of normal keratinocytes in the epidermis, which leads to the development of skin cancers including cSCC [2,4]. However, the detailed underlying molecular mechanisms for this transition still need to be fully elucidated.

Apoptosis is one type of programmed cell death, which is initiated by intrinsic or extrinsic signals and finely-tuned by many factors including BCL-2 family proteins [5]. BCL-2 family members, including both anti-apoptotic and pro-apoptotic BCL-2 family proteins, govern mitochondrial outer membrane permeabilization (MOMP) to regulate apoptosis. When apoptotic cell death program starts, MOMP releases cytochrome c to cytosol and activates caspase-family proteases for initiating apoptotic events. Tumors develop effective mechanisms to resist apoptosis and thus become one of the hallmark features of cancer [6]. The lead member BCL-2 and its homologue BCL-XL (BCL2L1) are featured by the presence of four conserved BH domains, three of which, BH1, BH2, and BH3 domains, are responsible for their antiapoptotic role mainly by interactions with other proapoptotic members of the BCL-2 family [7,8]. Inhibiting the activities of anti-apoptotic BCL-2 members by drugs [9] or neutralizing by BH3 peptides [10] to prime cancer cells to death has been regarded as effective therapeutic modalities. In another way, transcriptionally or post-transcriptionally repressing the expression levels of anti-apoptotic BCL-2 members in cancer is also promising. However, the mechanism regulating the expression of BCL-2 family members still remain unclear.

Autophagy is an evolutionarily-conserved catabolic process, during which “useless” proteins and damaged organelles are sequestrated into autophagosomes and fused with lysosomes to form autolysosomes for bulk degradation of embedded components [11]. Except the dysregulated autophagic process defined as type II programmed cell death, autophagy is generally regarded as protective cellular process [12]. The recycled peptides, nucleotides and lipids together with the energy are critical to the maintenance of cellular homeostasis and support tumor cells to survive under stressful environment [11]. Autophagy-related genes (ATGs), including ULK1 (ATG1), ATG3, and ATG12 etc., are involved in the distinct stages of autophagy: initiation, autophagosome nucleation, autophagosome membrane expansion, fusion with lysosome and the intravesicular components degradation [13]. Signaling pathways or critical factors are involved in regulating ATG genes, including Class III PI3-kinase complex I, mTORC I complex, mTORC II complex and Beclin1 et al. [14]. However, the transcriptional regulation of the ATG genes is not fully explored.

In our previous study, Homeobox A9 (HOXA9) was significantly downregulated and identified as tumor suppressor in cSCC tumors and cells. HOXA9 functions in repressing the proliferation, migration and invasiveness, while promoting apoptosis of cSCC cells [15]. However, the mechanisms about how HOXA9 regulate apoptosis has not been explored and remain to be elucidated. In this study, we found RELA (p65 subunit of NF-κB) is transcriptionally repressed by HOXA9. In cSCC, RELA transactivates *BCL-XL* to antagonize apoptosis and ATG genes (*ULK1*, *ATG3*, and *ATG12*) to promote autophagy. Our study highlights a HOXA9-NF-κB axis regulates both the apoptosis and autophagy to promote tumor development in cSCC, which may suggest novel intervention targets for cSCC therapy.

## 2. Materials and Methods

### 2.1. Animal Experiments

This research was agreed by the Institutional Animal Care and Use Committee (IACUC) of Nanfang Hospital (Code L2018240). Our performance follows the guidelines of the Asian Federation of Laboratory Animal Science Associations and the National Regulations for the Administration of Affairs Concerning Experimental Animals. Details of Mouse transportation, housing, and breeding were performed according to “The use of non-human primates in research”. Nude mice (BALB/C-nu/nu, 4 weeks old) for xenograft experiments were provided by the animal center of Southern Medical University. During sample collections, mice were sacrificed by cervical dislocation to prevent suffering.

### 2.2. Cell Line and Drug Treatment

cSCC line HSC-1 (Male, HonSun Biological, Shanghai, China) was cultured in Dulbecco’s modified Eagle medium (DMEM, Invitrogen, Carlsbad, CA, USA) supplemented with 10% fetal bovine serum (Serana, Aidenbach, Germany) and maintained at 37 °C with 5% CO_2_ in a humidified atmosphere. This cell line has been verified to be mycoplasma free. JSH-23 (SellckChem, Houston, TX, USA) was dissolved in DMSO and used for inhibition of NF-κB activity.

### 2.3. RNA Isolation and qRT-PCR

Total cellular RNAs were isolated by using TRIzol reagent (Life Technologies). cDNAs were prepared using Honor II 1st Strand cDNA Synthesis Supermix for qPCR (Novogene, Tianjin, China). mRNA expression levels were detected by Unique Aptamer Green Master Mix (Novogene, Tianjin, China) on LightCycler 96 Detection System (Roche, Basel, Switzerland). Detection of GAPDH mRNA expression was used for normalization. Primers used in this study are provided in Appendix A.

### 2.4. DNA Construct

We acquired HOXA9 expressing vector Addgene (http://www.addgene.org) and signed the material transfer agreement (MTA).

### 2.5. RNA-Seq

Total RNA Sequencing was done by Novogene using the Illumina HiSeq 2500. Tophat2 was used for the alignment (http://ccb.jhu.edu/software/tophat) and human assembly GRCh37/hg19 was used as the reference genome. Significant differentially-expressed genes were detected by HTSeq (http://www-huber.embl.de/HTSeq). On the website of DAVID Bioinformatics Resources 6.8 (http://david.abcc.ncifcrf.gov/), we performed Gene Ontology (GO) and Kyoto Encyclopedia of Genes and Genomes (KEGG) analyses of the differentially expressed genes.

### 2.6. Immunoblotting and Immunohistochemistry (IHC) Assays

Western blot was performed in accordance to standard protocol. The primary antibodies and dilutions used in this study were described as following: HOXA9 (Abcam, ab140631, 1:2000) and RELA (SellckChem, A5075, 1:2000); BCL-XL (SellckChem, Houston, TX, USA, A5091, 1:2000), ULK1 (Cell Signaling Technology, Danvers, MA, USA, #8054, 1:2000), ATG3 (SellckChem, A5304, 1:2000), ATG12 (SelleckChem, A5565, 1:2000) and GAPDH (Santa Cruz Biotechnology, sc-25778,1:5000). Anti-mouse IgG-horseradish peroxidase (HRP) and anti-rabbit IgG-HRP (Santa Cruz Biotechnology, Dallas, TX, USA) were used as secondary antibodies. Luminata Forte Western HRP substrate (Merck Millipore, Burlington, MA, USA) was used to detect the bound antibodies. Densimetric quantification of the Western bands was performed using Quantity One software (Bio-Rad, Philadelphia, PA, USA).

Xenograft tumors were formalin-fixed, paraffin-embedded and sectioned for IHC staining. The following antibodies were used: HOXA9 (Abcam; 1:100), RELA (1:100), BCL-XL (1:100), ULK1 (1:100), ATG3 (1:100) or ATG12 (1:100). Stained sections were imaged using BX53 microscope (Olympus) to get representative images for statistical analysis.

### 2.7. Xenograft Mouse Model

Briefly, male nude mice (BALB/C-nu/nu, 4 weeks old) were used in this study. 1.0 × 10^7^ cells were subcutaneously injected into the left and right flanks of the above nude mice. After implanted xenografts grows to around 100 mm^3^ in size, pre-incubated 3 nM siRNAs including siNC and siHOXA9 were respectively injected into the xenografts and the injection was repeated every three day. The handling persons were “blind” to the different treatments. Tumor growth curve was generated by measuring the tumor diameters every day. At the end of experiment (21 days after the first siRNA injection), the xenografts were dissected and snap-frozen immediately for later RNA and protein extractions or paraffin-embedded for IHC staining. 

### 2.8. ChIP-qPCR Analysis

Chromatin immunoprecipitation (ChIP) was done by using EZ ChIP Chromatin Immunoprecipitation kit (Cat. 17-371, Merck Millipore, Burlington, MA, USA). 5 μg antibodies against HOXA9 (Abcam, Cambridge, MA, USA) or isotype IgG (Merck Millipore, Burlington, MA, USA) was used. qRT-PCR was done using Unique Aptamer Green Master Mix (Novogene, Tianjin, China). Relative enrichment was calculated by normalizing to IgG immunoprecipitation. Primers used in this study are listed in Appendix A.

### 2.9. Cell Proliferation Assay

4000/well cells were seeded on 96-well plates and transfected with siRNAs. Cell proliferation was done after 0, 24, 48, and 72 h by cell counting kit (TransGen Biotech, Beijing, China) at 450 nm.

### 2.10. Apoptosis Assay

Cells on 60-mm dishes were transfected with siNC, siHOXA9_01, or siHOXA9_02 and cultivated for 36 h. Cell apoptosis was detected using TransDetect Annexin V-FITC/PI cell apoptosis detection kit (TransGen Biotech, Beijing, China) and quantified by flow cytometry using AQUIOS CL Flow Cytometer (Beckman Coulter, Brea, CA, USA).

### 2.11. Statistical Analysis

Statistical analysis was done using an unpaired *t*-test or one-way ANOVA tests (SPSS v20). All statistical tests incorporated two-tailed tests and homogeneity of variance tests and were thought to indicate significant difference if * *p* < 0.05, ** *p* < 0.01, or *** *p* < 0.001.

### 2.12. Data Availability

All data generated or analyzed in this study are included in this published article and its Appendix A, which are also available from the corresponding author on request.

## 3. Results

### 3.1. HOXA9 Is Predicted to Regulate Apoptotic- and Autophagic-Genes in cSCC

HOXA9 has been identified as a tumor suppressor in cSCC by inhibiting glycolysis while promoting apoptosis [15]. Yet, the exact roles of HOXA9 in regulating apoptosis process is still unclear. To understand how HOXA9 regulates the molecular events related with apoptosis and other cellular processes in cSCC cells, we repeated the bioinformatic analysis of the previous transcriptome sequencing after HOXA9 knockdown [15]. Gene Ontology (GO) analysis with the list of significantly up-regulated genes revealed that the top-ranked lists of enriched Gene Ontology categories includes “Positive regulation of apoptotic process”, “Apoptotic process”, “Regulation of apoptotic process”, “Regulation of extrinsic apoptotic signaling pathway via death domain receptors”, “Positive regulation of programmed cell death”, “Positive regulation of NF-kappaB transcription factor activity”, “Positive regulation of I-kappaB kinase/NF-kappaB signaling”, “Macroautophagy” and “Positive regulation of transcription, DNA-templated”, etc. (*p* < 0.05, Table 1, Appendix A). Markedly, Kyoto Encyclopedia of Genes and Genomes (KEGG) pathway analysis indicated that molecular signaling pathways including “NF-kappaB signaling pathway”, “Apoptosis” and “Autophagy” are significantly influenced (*p* < 0.05, Table 1, Appendix A). Among the genes affected in the above three pathways, significantly-upregulated genes including NF-κB, BCL2L1 (BCL-XL), ULK1 (ATG1), ATG3, and ATG12 were functionally relevant to apoptosis or autophagy.

### 3.2. HOXA9 Promotes Apoptosis While Inhibits Autophagy

To verify the apoptosis- and autophagy-regulatory roles of HOXA9 supported by the above bioinformatic analysis of transcriptomic sequencing, we performed assays to determine the variations of apoptosis after HOXA9 expression alterations. We did gain-of- and loss-of-functional tests and confirmed that depletion of HOXA9 enhances the proliferation and decreases the apoptosis of cSCC cells as shown by repressed cleaved CASPASE3 generation and apoptotic cells (Figure 1a–c), while overexpression of HOXA9 indeed plays tumor-suppressive roles by inhibiting the proliferation and promoting apoptosis in cSCC cells (Figure 1d–f), which is consistent with the previous report [15].

The status of autophagy in response to HOXA9 variation was also checked. Knockdown of HOXA9 enhances autophagy as shown by the increased LC3B II modification, P62 expression (Figure 1a) and LC3B immunofluorescence (Figure 1g), while overexpression of HOXA9 significantly inhibited autophagy (Figure 1d). Thus, we concluded that loss of HOXA9 inhibits apoptosis but promotes autophagy in cSCC cells.

### 3.3. HOXA9 Negatively Regulated the Expression of RELA, BCL-XL, ULK1, ATG3, and ATG12

To further validate the pro-apoptotic and anti-autophagic functions of HOXA9, *RELA* (encoding P65 subunit of NF-κB), *BCL-XL*, *ULK1*, *ATG3*, and *ATG12* were selected from the significantly varied genes owing to their critical roles in apoptosis and autophagy. qRT-PCR and western blot detection confirmed that the upregulation of RELA, BCL-XL, ULK1, ATG3, and ATG12 in response to HOXA9 knockdown (Figure 2a,b). Conversely, overexpression of HOXA9 inhibited the expression of RELA, BCL-XL, ULK1, ATG3, and ATG12 (Figure 2c,d). Collectively, HOXA9 regulates apoptosis and autophagy by negatively regulating anti-apoptotic and pro-autophagic genes.

### 3.4. HOXA9 Inhibits the Epigenetic Activities of NF-κB

HOXA9 is a known transcription factor functioning in transcriptionally regulating gene expression [16]. To explore how HOXA9 regulates the above apoptosis- and autophagy-related genes, the predicted promoter regions (+1000 bp to −1000 bp relative to the transcription start site [TSS]) of *RELA*, *BCL-XL*, *ULK1*, *ATG3*, and *ATG12* were analyzed according to a minimum consensus binding site (ATAA) via the UCSC genome browser (http://genome.ucsc.edu/) [17] and rVista (https://rvista.dcode.org/) [18]. Interestingly, HOXA9 binding site was only predicted to be present on the promoter of *RELA*, but not on *BCL-XL*, *ULK1*, *ATG3*, and *ATG12* loci (Figure 2e). Seeing the expression levels of *BCL-XL*, *ULK1*, *ATG3*, and *ATG12* are responsive to HOXA9 overexpression and knockdown, it promoted that they are potentially regulated by RELA. As anticipated, NF-κB banding site (aGGGgaTTTCCaxxx) was predicted to be presented on the promoter regions of *BCL-XL*, *ULK1*, *ATG3*, and *ATG12* genes (Figure 2e). Based on the above predictions, chromatin immunoprecipitation (ChIP)-qPCR detection was performed and confirmed that depletion of HOXA9 by RNA interference led to marked decrease in the binding enrichment of HOXA9 at the binding sites of *RELA* locus (Figure 2f) while the binding enrichments of RELA on *BCL-XL*, *ULK1*, *ATG3*, and *ATG12* loci are significantly enhanced (Figure 2f), which is consistent the upregulated expression of RELA, BCL-XL, ULK1, ATG3, and ATG12 after HOXA9 knockdown (Figure 2a,b). The above results clearly proved that HOXA9 negatively regulates BCL-XL, ULK1, ATG3, and ATG12 through transcriptionally-repressive regulation of RELA.

### 3.5. Inhibition of RELA Decreased the Expression of BCL-XL, ULK1, ATG3, and ATG12

To verify if HOXA9 knockdown could lead to increased NF-κB expression and enhanced NF-κB activity, IF staining of RELA protein was performed. Total RELA protein expression was strongly increased and enhanced RELA translocation to nucleus was apparently detected, which is comparable to TNF-α induction (Figure 3a). To validate if BCL-XL, ULK1, ATG3, and ATG12 are regulated by RELA, an inhibitor of NF-κB transcriptional activity, JSH-23, was used to treat cSCC cells. As predicted, JSH-23 treatment significantly repressed the expression of BCL-XL, ULK1, ATG3, and ATG12 in both mRNA and protein levels (Figure 3b,c). Interestingly, knockdown of HOXA9 could partially rescue RELA expression and thus promote BCL-XL and ATG genes expression in mRNA and protein levels, which verifies the expression level is critical for RELA in regulating downstream genes expression (Figure 3d,e). Thus, our results demonstrated that NF-κB directly promotes the expression levels of downstream *BCL-XL*, *ULK1*, *ATG3*, and *ATG12* genes.

### 3.6. Loss of HOXA9 Inhibits Apoptotic geNes While Enhances Autophagic Genes In Vivo

To evaluate the pro-apoptotic and anti-autophagic roles of HOXA9 in vivo, a xenograft tumor model was established in immunocompromised mice. Knockdown of HOXA9 strongly promoted tumor growth and the tumor sizes were apparently larger compared to siNC group at the end of the experimental period (Figure 4a). HOXA9 depletion in the siHOXA9 group was verified by western blot, qPCR and IHC staining (Figure 4b,c). Furthermore, HOXA9 depletion led to the significant up-regulation of RELA, BCL-XL, ULK1, ATG3, and ATG12 in both protein and RNA levels as shown in western blot, qPCR detection and IHC staining (Figure 4c–e). Collectively, our in vivo experiments clearly indicated that HOXA9 promotes apoptosis and represses autophagy by negatively regulating NF-κB and its downstream anti-apoptotic and pro-autophagic genes including *BCL-XL*, *ULK1*, *ATG3*, and *ATG12*.

## 4. Discussion

As one type of programmed cell deaths, apoptosis is characterized by a series of morphological changes including nuclear condensation, DNA fragmentation and formation of apoptotic bodies. Autophagy is generally regarded as a pro-survival cellular process by controlling the turnover of intracellular organelles and proteins. In most cases, autophagy stands as a survival strategy to deal with uncomfortable stresses [19]. Stresses often induce both of autophagy and apoptosis in the same cell, generally, autophagy appears earlier than apoptosis. However, if apoptosis is onset, autophagy will be inactivated [20]. The molecular mechanisms under such interplay between apoptosis and autophagy is debating and still under exploration [19]. 

HOXA9 belonging to HOX gene family is primarily identified to regulate embryonic development, maintain hematopoietic stem cells and play roles either as an oncogene or as a tumor suppressor in various tumors [21,22,23,24]. HOXA9 is previously identified to be significantly downregulated in cSCC, which contributes to the repressed apoptosis of cSCC cells [15]. In this study, bioinformatic analysis of the transcriptomic sequencing data after HOXA9 knockdown indicated that HOXA9 functions in regulating NF-κB pathway, apoptosis and autophagy (Table 1). Apoptosis-related genes including RELA and BCL-XL were significantly upregulated in response to HOXA9 depletion. Further study the upregulation of BCL-XL is not directly regulated by HOXA9, but by RELA, which also directly modulates the expression of ULK1, ATG3, and ATG12. Thus, RELA mediates the HOXA9′s regulation of BCL-XL, ULK1, ATG3. and ATG12.

The NF-κB family includes five DNA-binding proteins which can form homodimers or heterodimers and are critical regulators of cancer development [25]. Generally, upregulation or activation or of NF-κB enhance cell proliferation, inhibit apoptosis, stimulate cell migration and invasion [26]. RELA (p65) bind p50 to form heterodimer and mediates the transcriptional regulation of classical NF-κB signaling pathway to promote cell proliferation, survival, angiogenesis and metastasis [25]. Although HOXA9 has been reported to regulate RELA expression in Non-small Cell Lung Cancer, no report has demonstrated such regulation is direct or indirect [27]. Our work provides evidences to demonstrate the direct binding of HOXA9 on RELA promoter and deepen the understanding of HOXA-RELA regulator axis, which highlights the tumor-suppressive role of HOXA9 in cSCC.

The BCL-2 family includes important apoptosis regulators. The balanced expression between pro-apoptotic members (BAX, BAK, BAD, BIK and BID) and antiapoptotic members (BCL-2, BCL-XL, BCL-W and MCL-1) maintains the homeostasis and survival of cells [28]. Disruption of the above balance, especially the repression of the antiapoptotic members, will lead to apoptosis. BCL-XL contains conserved topology of BCL-2 family and binds with pro-apoptotic ligands to promote pro-survival features [29]. The regulation of BCL-XL by NF-κB has been reported by previous reports [30,31]. In this study, we further identified the transcriptional regulatory binding site that the NF-κB p65 subunit, RELA, directly binds and promote the expression of BCL-XL. ULK1 (ATG1) is critical for the initiation of autophagy by perceiving the nutrient-related cellular status, recruiting ATG proteins to form autophagosomes [32]. ATG3 is an E2-like conjugating enzyme mediating LC3 lipidation while acts as alternative conjugate of ATG12. ATG12-ATG3 complex not only regulates basal autophagy, also connects the late endolysosomal trafficking [33].

As BCL-2 family proteins and ATG proteins are critical factors for apoptosis or autophagy respectively, their turnover mechanisms are of profoundly interest. Also, one open question is how the two critical survival processes, apoptosis and autophagy, are connected and balanced by any molecular switcher or regulator? The HOXA9-NF-κB regulatory axis identified demonstrates how one tumor suppressor, HOXA9, negatively enhances apoptosis and suppresses autophagy through transcriptionally regulating NF-κB. Regulating both apoptotic and autophagic factors at the same time by NF-κB is an important novel finding in this study, which primarily give the answer for the above question.

Collectively, our findings establish a novel HOXA9-NF-κB regulatory axis that functions in regulating NF-κB-mediated apoptosis and autophagy in cSCC. As depicted in Figure 4f, HOXA9 is down-regulated in both cSCC cells [15]. Loss of HOXA9 up-regulates NF-κB and its downstream genes including *BCL-XL*, *ULK1*, *ATG3*, and *ATG12*, which regulate apoptotic and autophagic pathway and contributes to the survival of cSCC cell and cSCC progression. HOXA9 directly bind the promoter and negatively regulates the expression of NF-κB is critical for HOXA9′s indirect modulation of apoptotic and autophagic pathways. As the mediator, NF-κB directly binds the promoters of *BCL-XL*, *ULK1*, *ATG3*, and *ATG12* genes to inhibit apoptosis and enhance autophagy at the same time, which executes the pro-survival function. Our findings highlight the pro-apoptotic and anti-autophagic roles of HOXA9 in skin tissue, support our previous finding that HOXA9 acts as tumor suppressor in cSCC, and emphasize a newly identified HOXA9- NF-κB axis that may provide novel intervention targets for cSCC therapy.

## Figures and Tables

**Figure 1 cells-08-01360-f001:**
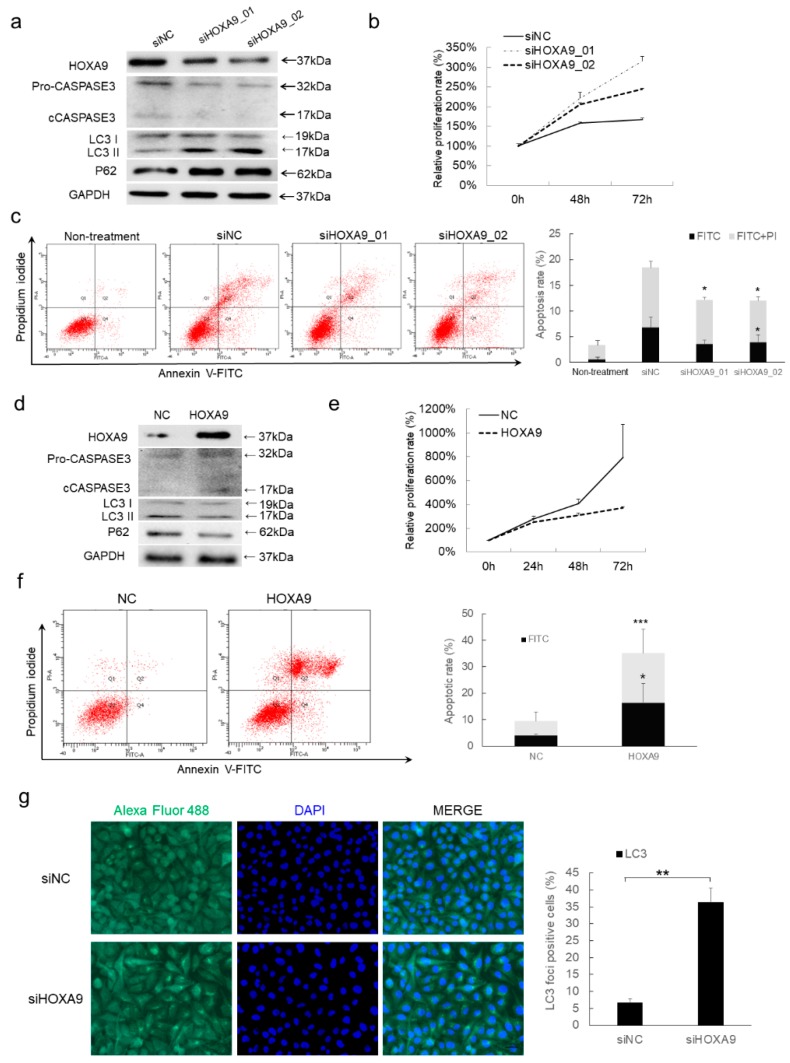
HOXA9 represses cell proliferation while promotes apoptosis in cSCC cells. (**a**) HOXA9 protein expression was detected after knockdown of HOXA9 by siRNAs in cSCC cells. Measurements of cell proliferation by CCK-8 assay (*n* = 3) (**b**) and apoptosis assay by Annexin V/PI double staining (*n* = 3) (**c**) were performed in cSCC cells treated with siRNAs targeting HOXA9. (**d**) HOXA9 protein expression was detected by western blot after overexpression of HOXA9 in cSCC cells. Measurements of cell proliferation by CCK-8 assay (*n* = 3) (**e**) and apoptosis assay by Annexin V/PI double staining (*n* = 3) (**f**) were performed in cSCC cells overexpressing HOXA9. (**g**) The autophagy in cSCC cells following HOXA9 knockdown was evaluated by LC3 staining. * *p* < 0.05, ** *p* < 0.01, *** *p* < 0.001.

**Figure 2 cells-08-01360-f002:**
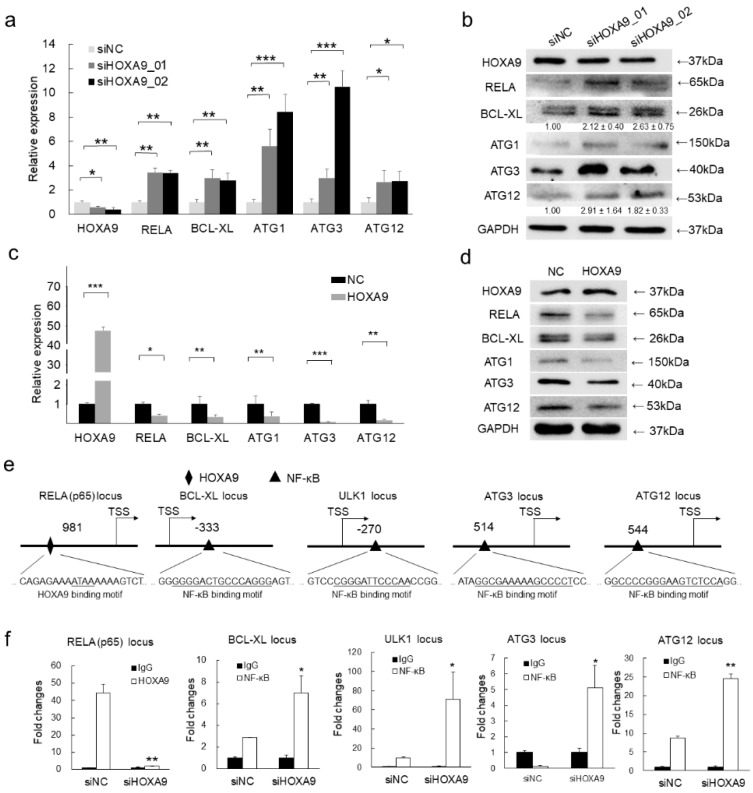
HOXA9 represses the expression of NF-κB and its downstream apoptotic and autophagic genes by directly binding to the promoter region of NF-κB. (**a**–**d**) The mRNA or protein expression levels of HOXA9, RELA (p65), BCL-XL, ULK1, ATG3, and ATG12 were detected by qRT-PCR (*n* = 3) or western blot in cSCC cells after knockdown of HOXA9 by two siRNAs or overexpression of HOXA9. The qRT-PCR data were normalized to GAPDH gene expression. In western blots, GAPDH was used as a loading control. The bands of BCL-XL and ATG12 were densimetrically quantified (*n* = 3). (**e**) Predicted binding site of HOXA9 (diamond) at the promoter of RELA (p65) or binding sites of RELA for on the promoter regions of *BCL-XL*, *ULK1*, *ATG3*, and *ATG12* by rVista (https://rvista.dcode.org/). (**f**) The binding enrichment of HOXA9 at the *RELA* locus or RELA at the loci of *BCL-XL*, *ULK1*, *ATG3*, and *ATG12* was detected by ChIP-qPCR after knockdown of HOXA9. One-Way ANOVA and Dunnett’s multiple comparison test. Means ± s.d., * *p* < 0.05, ** *p* < 0.01, *** *p* < 0.001.

**Figure 3 cells-08-01360-f003:**
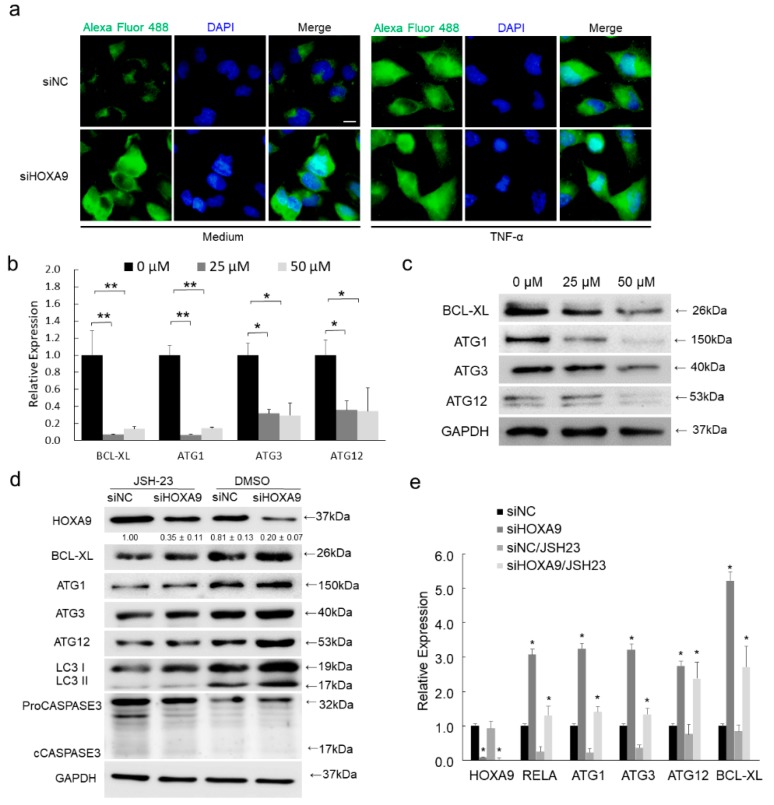
Loss of HOXA9 enhanced NF-κB activity while inhibition of NF-κB decreased the expression of BCL-XL, ULK1, ATG3, and ATG12. (**a**) Total RELA protein expression and RELA translocation to nucleus was detected by IF staining. (**b**,**c**) The expression levels of BCL-XL, ULK1, ATG3, and ATG12 in both mRNA and protein levels were detected after the treatment with NF-κB inhibitor JSH-23 using 25 or 50 μM. (**d**,**e**) JSH-23 treatment was performed after HOXA9 knockdown to verify the effects of RELA in regulating the mRNA and protein expression levels of BCL-XL, ULK1, ATG3, and ATG12. The bands of HOXA9 were densimetrically quantified (*n* = 3). * *p* < 0.05, ** *p* < 0.01.

**Figure 4 cells-08-01360-f004:**
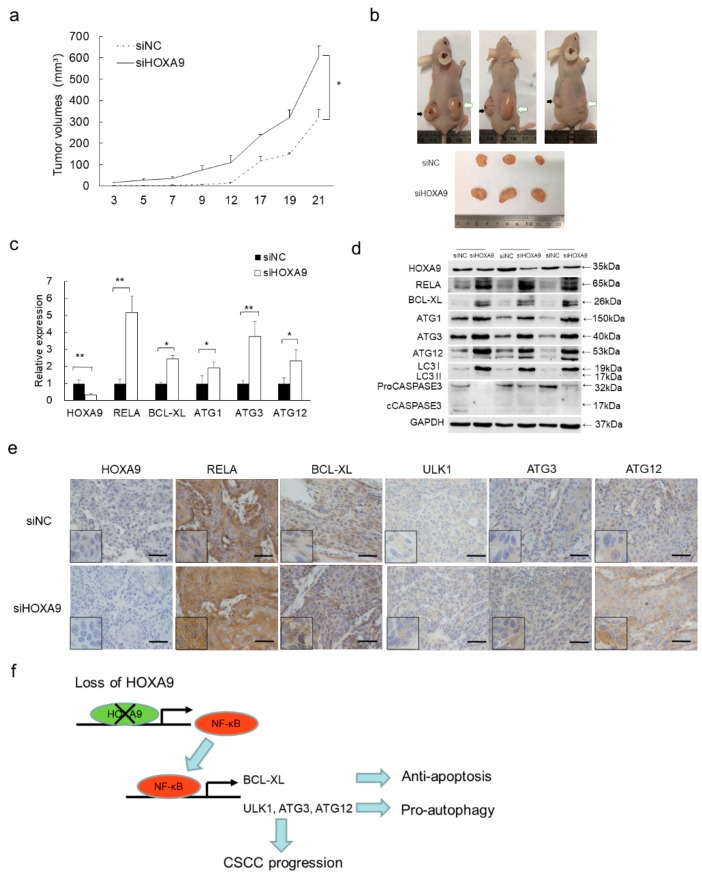
Loss of HOXA9 inhibited apoptosis, promoted autophagy and tumor growth in vivo. (**a**) HOXA9 depletion enhances xenografts growth. Statistical data of tumor volumes represent the average of three independent experiments ± s.d, respectively. (**b**) Dissected xenografts from three sacrificed mice were imaged at the end of experiment. Black arrows point the siNC-treated xenografts while white arrows indicate siHOXA9-treated xenografts. (**c**) The expression of *HOXA9*, *RELA*, *BCL-XL*, *ULK1*, *ATG3*, and *ATG12* was detected in the dissected xenografts by qRT-PCR. Statistical data of qRT-PCR represent the average of three independent experiments ± s.d. (**d**) The protein expression levels of HOXA9, RELA (p65), BCL-XL, ULK1, ATG3, and ATG12 was detected in xenografts after siHOXA9 treatment by western blot. (**e**) Histopathology analysis (IHC staining) of HOXA9, RELA (p65), BCL-XL, ULK1, ATG3, and ATG12 on tumor sections. Scale bar, 100 µm. (**f**) A model of the HOXA9-NF-κB regulatory axis in cSCC development. In cSCC tumors, loss of HOXA9 up-regulates NF-κB and its downstream anti-apoptotic gene *BCL-XL* and pro-autophagic genes of *ULK1*, *ATG3*, and *ATG12*, which contributes to the repressed apoptosis, enhanced autophagy and promotes cSCC progression. * *p* < 0.05, ** *p* < 0.01.

**Table 1 cells-08-01360-t001:** Transcriptomic analysis of HOXA9-regulated genes by RNA-Seq in cutaneous squamous cell carcinoma (cSCC) cells. Over-represented categories by GO (Gene Ontology) and KEGG (Kyoto Encyclopedia of Genes and Genomes) pathway analysis of differently-expressed genes. BP: biological process. The “NF-kappaB signaling pathway”, “Apoptosis” and “Regulation of autophagy” were highlighted.

Category	GO ID	Term	*p*-Value
GOTERM_BP_DIRECT	GO:0045893	Positive regulation of transcription, DNA-templated	5.65 × 10^−12^
GOTERM_BP_DIRECT	GO:0043065	Positive regulation of apoptotic process	1.58 × 10^−9^
GOTERM_BP_DIRECT	GO:0045944	Positive regulation of transcription from RNA polymerase II promoter	2.14 × 10^−8^
GOTERM_BP_DIRECT	GO:0051092	Positive regulation of NF-kappaB transcription factor activity	2.19 × 10^−8^
GOTERM_BP_DIRECT	GO:0006915	Apoptotic process	4.17 × 10^−8^
GOTERM_BP_DIRECT	GO:0043123	Positive regulation of I-kappaB kinase/NF-kappaB signaling	5.28 × 10^−8^
GOTERM_BP_DIRECT	GO:0042981	Regulation of apoptotic process	5.16 × 10^−6^
GOTERM_BP_DIRECT	GO:1902041	Regulation of extrinsic apoptotic signaling pathway via death domain receptors	1.12 × 10^−4^
GOTERM_BP_DIRECT	GO:0043068	Positive regulation of programmed cell death	1.57 × 10^−3^
GOTERM_BP_DIRECT	GO:0016236	Macroautophagy	1.78 × 10^−2^
KEGG_PATHWAY	hsa05200	Pathways in cancer	6.91 × 10^−12^
KEGG_PATHWAY	hsa04064	NF-kappaB signaling pathway	2.63 × 10^−9^
KEGG_PATHWAY	hsa04210	Apoptosis	1.08 × 10^−2^
KEGG_PATHWAY	hsa04140	Regulation of autophagy	4.96 × 10^−2^

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
