# Peer review of "HOXA9 Transcriptionally Promotes Apoptosis and Represses Autophagy by Targeting NF-κB in Cutaneous Squamous Cell Carcinoma"

_cells, 2019, doi:10.3390/cells8111360_

Round 1

Reviewer 1 Report

In this manuscript, Han et al. demonstrated the tumor suppressing role of HOXA9 in cutaneous squamous cell carcinoma (cSCC) cells. Using the gain and loss of HOXA9 gene, they revealed that HOXA9 induced apoptosis, but suppressed autophagy in cSCC cells by transcriptionally regulating the p65 subunit of NF-kB (Rel A). RelA transcriptionally promoted the anti-apoptotic protein BCL-XL as well as autophagic factors ATG1, ATG3 and ATG12. They also confirmed that the loss of HOXA9 repressed tumor apoptosis with enhanced autophagy using a xenograft tumor model of cSCC.

Although the manuscript presented novel findings, but it needs to be largely revised. The authors could not properly explain the autophagic regulation of HOXA9. I have several major concerns which need to be addressed to improve the manuscript for publication.

In 2b, the labeling of Y axis is missing. They should show the % of cell viable. Control cells viability should be 100%. In figure 2, why are so much cell death (apoptosis+necrosis) under siNC (control)? More cells are in PI+AnnV in siNC group. They should present without any siRNA transfection as control. Are this cells dying without any treatment with in 48 hours of plating? p62/SQSTM1 is useful as a marker of autophagic defect in cells and tissues. The increase in p62 in tissues indicates the possibility of an insufficiency in autophagy.

Therefore, please explain, in Figure 2a and d, why are both LC3II and p62 increase? They should be inversely regulated. Please explain it.

It is important to examine the translocation of the transcription factor, NFkB (p65 and p50), from cytosol to nucleus. Not only total p65 level as protein or RNA Figure legend for figure 3 g and h are missing. They should indicate JSH-23 with different concentrations in this figure. g and h. In figure 2a, the authors showed that siHOXA9 increased LC3II than siNC. But in Figure 4 of tumor study, they showed the opposite regulation of LC3II with siHOXA9. Please explain. Autophagy is regulated by mTOR. Authors should examine the regulatory role of HOXA9 on mTOR activity. “Tumor volume statistical data represent the average of three independent experiments ± s.d, respectively”. Did they use 3 animals per group? N=3 is very low number for in vivo experiment. All other experiments, please indicate n value in each figure. In xenograft mouse model “1.0 × 107 cells were subcutaneously implanted into the left and right flanks of nude mice. After implantation, siNC or siHOXA9 oligos were injected into the left or right tumor, respectively; and the injection was repeated every three day.” After how many days of implantation of cells, siRNA were injected? How long was siRNA injected? 21 days? These are not clear in Method section and in figure legend. Figure 4e, the immunohistochemistry pictures are not impressive.

Author Response

Reviewer 1:

Q: In 2b, the labeling of Y axis is missing. They should show the % of cell viable. Control cells viability should be 100%.

A: Thanks for Reviewer’s comments. The labeling of Y axis has been added and the curves have been redraw to meet the requirements (Figure 2b,e).

Q: In figure 2, why are so much cell death (apoptosis+necrosis) under siNC (control)? More cells are in PI+AnnV in siNC group. They should present without any siRNA transfection as control. Are this cells dying without any treatment within 48 hours of plating?

A: Thanks for Reviewer’s comments. We include the image showing the apoptosis status without any siRNA transfection as control. Generally, the academia acknowledged that the cell death appeared in siNC is method-induced which is caused by siRNA transfection. From image showing the apoptosis status without any siRNA transfection, we can clearly observe that there is little cell death without any treatment within 48 hours of plating. In this situation, knockdown of HOXA9 clearly shows the anti-apoptotic effects.

Q: p62/SQSTM1 is useful as a marker of autophagic defect in cells and tissues. The increase in p62 in tissues indicates the possibility of an insufficiency in autophagy. Therefore, please explain, in Figure 2a and d, why are both LC3II and p62 increase? They should be inversely regulated. Please explain it.

A: Thanks for Reviewer’s comments. Actually, we also feel interesting when we observe the increased expression of P62 after HOXA9 knockdown. As Reviewer said, enhanced autophagy will generally lead to the degradation of P62 protein if the autophagy flux goes well. However, this is not always like the classical model and is actually context-dependent in practice, e.g. the sufficient autophagy was not always accompanied by the decreased expression of P62 protein. At least, two possibilities may explain for this result. Firstly, HOXA9 is a transcription factor and if P62 is directly or indirectly regulated by HOXA9 is unknown. Because P62 did not appear in the significant list after HOXA9 knockdown, we thus have not explored for this. Secondly, p62 is involved in proteasomal pathway, and the protein level may also increase when the proteasome degradation is inhibited. If any factor plays role in proteasome pathway is under the direct transcriptional regulation by HOXA9, the protein level of P62 may not fit the classical model. Previous studies already reported such situation (Bardag-Gorce F, et al., Modifications in p62 occur due to proteasome inhibition in alcoholic liver disease. Life Sci 2005; 77:2594-2602). Here, we just want to present these interesting results for reference or future exploration. If Reviewer and/or Editor think it is unsuitable, we may remove it from Figure 2.

Q: It is important to examine the translocation of the transcription factor, NFkB (p65 and p50), from cytosol to nucleus. Not only total p65 level as protein or RNA.

A: Thanks for Reviewer’s comments. We have done immunofluorescence staining to check the translocation status of NF-κB. As shown in Figure 4a, knockdown of HOXA9 indeed not only increase the overall amount of NF-κB but also promoted the translocation of NF-κB from cytosol to nucleus. Thus, the enhanced effects of NF-κB after HOXA9 knockdown is owing to the increased NF-κB protein expression which contributes to the increased amount of activated NF-κB.

Q: Figure legend for figure 3 g and h are missing. They should indicate JSH-23 with different concentrations in this figure. g and h.

A: Thanks for Reviewer’s comments. The figure legend for figure 3g and h have been added and the concentrations of JSH-23 have been indicated in this figure.

Q: In figure 2a, the authors showed that siHOXA9 increased LC3II than siNC. But in Figure 4 of tumor study, they showed the opposite regulation of LC3II with siHOXA9. Please explain.

A: Thanks for Reviewer’s comments. In Figure 4, the amount of LC3II is indeed increase when knockdown of HOXA9 expression. It is just because the LC3II signal is not that strong compared with LC3I band.

Q: Autophagy is regulated by mTOR. Authors should examine the regulatory role of HOXA9 on mTOR activity.

A: Thanks for Reviewer’s comments. We understand mTOR often plays roles in regulating autophagy. However, we have checked several times that the expression or activity of mTOR is definitely not regulated by HOXA9 in whatever depletion or overexpression of HOXA9. Seeing such results, we thus do further exploration in our RNA-seq data and discovered the critical roles of NF-κB in regulating the apoptosis and autophagy.

Q: Tumor volume statistical data represent the average of three independent experiments ± s.d, respectively”. Did they use 3 animals per group? N=3 is very low number for in vivo experiment. All other experiments, please indicate n value in each figure.

A: Thanks for Reviewer’s comments. Yes, we use 3 mice per group. As required, we have added the n value for each figure.

Q: In xenograft mouse model “1.0 × 107 cells were subcutaneously implanted into the left and right flanks of nude mice. After implantation, siNC or siHOXA9 oligos were injected into the left or right tumor, respectively; and the injection was repeated every three day.” After how many days of implantation of cells, siRNA were injected? How long was siRNA injected? 21 days? These are not clear in Method section and in figure legend.

A: Thanks for Reviewer’s comments. We have added more details to the “2.7 Xenograft Mouse Model” section to meet the requirements.

Q: Figure 4e, the immunohistochemistry pictures are not impressive.

A: Thanks for Reviewer’s comments. I guess this impression is due to the size of figures because we need to provide enough information in one figure. We guarantee the original figure is very impressive to show the differences.

Reviewer 2 Report

   In the manuscript by Han et al, the authors study the pro-apoptotic role of HoxA9 in squamous cell carcinoma, one of the most common forms of skin cancer. To understand better the mechanism behind this effect, the authors analyze previous RNAseq expression studies in HOXA9-downregulated SCC cells published by the corresponding author (Zhou eta al 2018). Among the factors identified by the authors are the anti-apoptotic factor BCL-XL, Nf-kB and autophagy-related genes.

The authors show that loss or overexpression of HOXA9 has an opposed effect on RelA and the rest of genes and propose a model where the role of HOXA9 as a pro-apoptotic and autophagy factor in SCC could be caused by the downregulation of NF-kb, which in turn would allow the expression of BCL-XL and the expression of autophagy-related genes.

The work is potentially interesting but has several major problems. The first main issue is novelty, because a previous report already suggested an inhibitory effect of HOXA9 over Nf-kB signaling in SCC (Yu et al, Mol Carcinog 2016). This is not mentioned or discussed anywhere in the text. Considering these evidences, to make the claims of the current relevant enough, the authors should characterize further this link between HOXA9 and NF-kB. However that was not the case. Although the authors claim that the effect of HoxA9 in apoptosis and autophagy is due to hyperactivation of the pathway, the authors have not demonstrated that NF-kB is functionally relevant there. The presence of NF-kB in few genes by ChIP cannot be claimed as a demonstration. The authors need to show that modulation of Nf-kB can revert functionally the impact of HOXA9 up/downregulation. If demonstrated properly at the cellular level and in xenografts, this would increase considerably the relevance of the work.

Some options to address this issue could be to inhibit the pathway by downregulating RelA o by chemical inhibition of the pathway (JSH-23) in the HOXA9-downregulated SCC cells. Another one could be to activate the pathway (overexpression of specific factors, incubation with cytokines like TNFa) in the HoxA9 overexpressed cells. In any of these cases, would the modulation of NF-kB revert the effect of HoXA9 downregulation or overexpression in proliferation, apoptosis and in tumorigenesis? I think this is a key issue and should be addressed by the authors.

Other major issues I have:

The starting point of the work is a RNAseq previously published by the last author in Nat comm where they claimed that HOXA9 inhibits glycolysis through a decrease in HIF1a signaling, which has a major impact in SCC growth and development. How do the authors reconcile both studies? An interplay between HIF1a and NF-kB has been proposed by several studies, so is possible that the NF-kB effect is due to this inhibition of HIF1a pathway?

The work has some technical problems: One of the most surprising observations is the considerable effect induced by a very mild downregulation of HoxA9. This shouldn’t be in principle a problem if everything fits, but there are a couple of issues that are problematic and should be addressed by the authors. First, How do the authors explain that in fig 3f (first ChIP), HoxA9 downregulation, which decreases HoxA9 in roughly 50-60% (or even less If we consider the GAPDH loading control in Fig 2a), completely abrogates HoxA9 localization to RELA gene?. Second, in Figure 3b both siRNas do not seem to have similar effects on ATG3 or RelA. How do the authors explain this discrepancy? Linked to this, why do the authors use siRNA # 2 and 3 in the RT-PCRs of Fig 3a and #1 and 2 in 3b? Is there any logical explanation for that? I cannot see a clear significant difference in the WB of fig 3b in the levels of BCL-XL or ATG12 particularly when the levels of the loading control of GAPDH are not equal in lane 1 vs 2-3 (there seem to be more in 2 and 3). If the authors are right, then I would recommend including a quantification of several WB to make their point. This issue should be addressed by the authors.

The work has many formal issues. Some of them: Many references lack the name of the journal (one of the cases is this Nat comm mentioned above, (Ref 15, Zhou et al, 2018). Graphs in fig 2b,c,e and f lack y-axis legend. Besides, the figure legends of these graphs have almost no information. Figure legends of Fig 3g,h are missing.

Author Response

Reviewer 2

Q: The work is potentially interesting but has several major problems. The first main issue is novelty, because a previous report already suggested an inhibitory effect of HOXA9 over Nf-kB signaling in SCC (Yu et al, Mol Carcinog 2016). This is not mentioned or discussed anywhere in the text.

A: Thanks for Reviewer’s comments. The mentioned reference has been discussed in the Discussion section.

Q: Considering these evidences, to make the claims of the current relevant enough, the authors should characterize further this link between HOXA9 and NF-kB. However that was not the case. Although the authors claim that the effect of HoxA9 in apoptosis and autophagy is due to hyperactivation of the pathway, the authors have not demonstrated that NF-kB is functionally relevant there. The presence of NF-kB in few genes by ChIP cannot be claimed as a demonstration. The authors need to show that modulation of Nf-kB can revert functionally the impact of HOXA9 up/downregulation. If demonstrated properly at the cellular level and in xenografts, this would increase considerably the relevance of the work. Some options to address this issue could be to inhibit the pathway by downregulating RelA o by chemical inhibition of the pathway (JSH-23) in the HOXA9-downregulated SCC cells. Another one could be to activate the pathway (overexpression of specific factors, incubation with cytokines like TNFa) in the HoxA9 overexpressed cells. In any of these cases, would the modulation of NF-kB revert the effect of HoXA9 downregulation or overexpression in proliferation, apoptosis and in tumorigenesis? I think this is a key issue and should be addressed by the authors.

A: Thanks for Reviewer’s guidance. We have done experiment to validate the if modulation of NF-κB can revert functionally the impact of HOXA9 up/downregulation? After experimental validation, we are glad by observing that treatment of cSCC cells with JSH-23 could revert the up-regulatory effects of BCL-XL and ATG genes induced by HOXA9 knockdown (Figure 4d). For xenografts verification, we have previously tried to use JSH-23 in animal studies. However, this inhibitor looks not only to function locally but spread to the whole body and show global effects, which led to several abnormalities in mice. We suggest to leave this xenograft verification in our future study to systematically resolve it.

Q: The starting point of the work is a RNAseq previously published by the last author in Nat comm where they claimed that HOXA9 inhibits glycolysis through a decrease in HIF1a signaling, which has a major impact in SCC growth and development. How do the authors reconcile both studies? An interplay between HIF1a and NF-kB has been proposed by several studies, so is possible that the NF-kB effect is due to this inhibition of HIF1a pathway?

A: Thanks for Reviewer’s suggestion. As the Reviewer said and what we have demonstrated that HIF-1α signaling plays a major impact in SCC growth and development. Now, this study validated the pro-apoptosis and anti-autophagy roles of HOXA9 through regulating NF-κB. The two studies are apparently different while tightly relevant. Crosstalk among different pathways is common. We also suppose an interplay between HIF-1α and NF-κB may also exist in cSCC as demonstrated by other studies. Yet, we believe future systemic investigation should be performed in cSCC before we draw conclusions.

Q: The work has some technical problems: One of the most surprising observations is the considerable effect induced by a very mild downregulation of HoxA9. This shouldn’t be in principle a problem if everything fits, but there are a couple of issues that are problematic and should be addressed by the authors. First, How do the authors explain that in fig 3f (first ChIP), HoxA9 downregulation, which decreases HoxA9 in roughly 50-60% (or even less If we consider the GAPDH loading control in Fig 2a), completely abrogates HoxA9 localization to RELA gene?

A: Thanks for editor’s comments. We also guessed that even knockdown removed most of HOXA9 protein, the residual HOXA9 protein should be functioning. However, our observations show it completely abrogates the HOXA9 binding on RELA locus. Generally, the binding of a transcription factor to promoter or enhancer region is competed by both positively and negative regulators. The binding proportion of HOXA9 on RELA locus is not linearly correlated with its amount within cells. For this interesting question, we guess when the total amount of HOXA9 decreases to a threshold like 40-50% in this study, the binding ability of HOXA9 may not be competitive and loses very quickly to even no detectable binding enrichment.

Q: Second, in Figure 3b both siRNas do not seem to have similar effects on ATG3 or RelA. How do the authors explain this discrepancy? Linked to this, why do the authors use siRNA # 2 and 3 in the RT-PCRs of Fig 3a and #1 and 2 in 3b? Is there any logical explanation for that? I cannot see a clear significant difference in the WB of fig 3b in the levels of BCL-XL or ATG12 particularly when the levels of the loading control of GAPDH are not equal in lane 1 vs 2-3 (there seem to be more in 2 and 3). If the authors are right, then I would recommend including a quantification of several WB to make their point. This issue should be addressed by the authors.

A: Thanks for editor’s comments. For Figure 3b, knockdowns of HOXA9 with both of the siRNAs increase the expression of ATG3 and RELA and the differences are just in folds. This is common because ATG3 and RELA are different genes and their expression levels are regulated not only in transcriptional level but also in post-transcriptional levels. There are only two siRNAs, siHOXA_001 and siHOXA9_002, were used in this study. The appearance of siRNA #2 and 3 in Fig. 3a and 3b is typo error and we are very sorry about this careless mistake. For the mentioned BCL-XL and ATG12 bands with insignificant differences, we did quantification to show the differences of protein expression levels to meet the requirement.

Q: The work has many formal issues. Some of them: Many references lack the name of the journal (one of the cases is this Nat comm mentioned above, (Ref 15, Zhou et al, 2018). Graphs in fig 2b,c,e and f lack y-axis legend. Besides, the figure legends of these graphs have almost no information. Figure legends of Fig 3g,h are missing.

A: Thanks for Reviewer’s comments. Ref 15 has been corrected as required. The Y-axis legends have been added as required. The missed figure legends of Fig 3g,h (now Fig. 4b,c) have been added as required.

Round 2

Reviewer 2 Report

   In the new version of the manuscript by Han et al, the authors have answered some of my concerns, but there are two issues that have not been properly addressed. The first one is regarding my major issue with the work, the lack of evidence demonstrating that the direct effect of HOXA9 on NF-kB signaling. The authors have performed an experiment in SCC cells combining HOXA9 downregulation with NF-kB chemical inhibition and testing the protein levels of common targets (Figure 4d). This experiment is very important and if properly shown, could be key to strengthen the main claim of the authors. However, in its current state, it’s far from convincing. The major issue is that in lanes 1 and 2, where the NF-kB inhibitor is used, no clear downregulation of HOXA9 is observed, particularly If one analyzes the levels of loading control GAPDH. Looking in detail, the very mild decrease of HOXA9 in lane 2 vs 1 is quite similar to the differences in GAPDH. This experiment should be properly repeated and quantified. Furthermore, since the main role of both proteins is to regulate gene expression, I would suggest to include also the RNA expression levels of these genes measured by qPCR.

The second issue is related to the problems with BCL-XL and ATG12 western-blots in Figure 3b. The authors have included numbers below both western-blots, but this is not enough to be convincing. They should show a quantification of n=3 Western-blots and statistics. It would not need to be included in the main figures, but it should be included to be able to make these claims.

Author Response

Comments from Reviewer 2

In the new version of the manuscript by Han et al, the authors have answered some of my concerns, but there are two issues that have not been properly addressed. The first one is regarding my major issue with the work, the lack of evidence demonstrating that the direct effect of HOXA9 on NF-kB signaling. The authors have performed an experiment in SCC cells combining HOXA9 downregulation with NF-kB chemical inhibition and testing the protein levels of common targets (Figure 4d). This experiment is very important and if properly shown, could be key to strengthen the main claim of the authors. However, in its current state, it’s far from convincing. The major issue is that in lanes 1 and 2, where the NF-kB inhibitor is used, no clear downregulation of HOXA9 is observed, particularly If one analyzes the levels of loading control GAPDH. Looking in detail, the very mild decrease of HOXA9 in lane 2 vs 1 is quite similar to the differences in GAPDH. This experiment should be properly repeated and quantified. Furthermore, since the main role of both proteins is to regulate gene expression, I would suggest to include also the RNA expression levels of these genes measured by qPCR.

A: Thanks for Reviewer’s suggestion. We have repeated this experiment and quantified both of the mRNA and protein expression levels. The new results have been updated to Figure 4d.

The second issue is related to the problems with BCL-XL and ATG12 western-blots in Figure 3b. The authors have included numbers below both western-blots, but this is not enough to be convincing. They should show a quantification of n=3 Western-blots and statistics. It would not need to be included in the main figures, but it should be included to be able to make these claims.

A: Thanks for Reviewer’s comments. The quantification (n=3) of BCL-XL and ATG12 western blots have been done and updated to Figure 3b.